# The Impact of Diet and Sociodemographic Factors on Cardiovascular Health in University Students

**DOI:** 10.3390/ijerph22050752

**Published:** 2025-05-10

**Authors:** María Victoria Padilla Samaniego, Angélica María Solís Manzano, Verónica Patricia Sandoval Tamayo, Edgar Rolando Morales Caluña, Katherine Denisse Suárez González, Nathalia Solórzano Ibarra

**Affiliations:** 1Research Group in Nutrition, Dietetics, Biotechnology and Food Analysis, State University of Milagro, Milagro 091050, Ecuador; asolism2@unemi.edu.ec (A.M.S.M.); vsandovalt@unemi.edu.ec (V.P.S.T.); 2Faculty of Health Sciences, State University of Milagro, Milagro 091050, Ecuador; emoralesc4@unemi.edu.ec (E.R.M.C.); ksuarezg@unemi.edu.ec (K.D.S.G.); nsolorzanoi@unemi.edu.ec (N.S.I.)

**Keywords:** balanced diet, socio-demographic factors, cardiovascular health, university students, dietary intake, triglyceride levels

## Abstract

This study aimed to analyze the association between diet, sociodemographic factors, and cardiovascular health in public university students, through a cross-sectional study conducted at the State University of Milagro, Ecuador, during 2022–2023. A total of 204 students participated, with demographic and health data collected through interviews and clinical measurements. The results showed that 22% of participants had a very high fat intake, while only 2% consumed a nutrient-rich diet. In women, a high-fat diet was associated with elevated triglyceride levels and higher systolic and diastolic blood pressure, whereas in men, it was linked to higher systolic and diastolic blood pressure. Conversely, a nutrient-rich diet was associated with lower systolic blood pressure in women. Place of birth significantly influenced systolic blood pressure in women and LDL cholesterol and diastolic blood pressure levels in men, with men born in Milagro showing higher LDL cholesterol and diastolic blood pressure compared with those born elsewhere. Additionally, place of residence was associated with systolic blood pressure in men. This study highlights the importance of promoting a balanced diet and considering sociodemographic factors when designing public health interventions to improve the cardiovascular health of university students.

## 1. Introduction

Cardiovascular diseases remain a global health concern due to their high prevalence and persistent consequences, as highlighted by the World Health Organization. Approximately 17.9 million people die annually from cardiovascular diseases, representing 31% of all deaths worldwide [1,2]. In recent years, there has been an observed increase in the prevalence of cardiovascular disease among young adults. This trend has been linked to declining quality of diet, which elevates the risk of conditions such as coronary artery disease, central obesity, increased blood glucose levels, and higher body mass index (BMI), all contributing to increased risk of stroke [3,4].

Despite substantial research on cardiovascular diseases [5,6], few studies have focused on the combined impact of diet and socio-demographic factors on cardiovascular health, particularly among young adults. University students, as a transitional population group, face unique challenges in maintaining healthy dietary habits owing to lifestyle changes, stress, and environmental factors [7]. This gap in the literature highlights the need for studies that explore how dietary patterns and socio-demographic contexts interact to influence cardiovascular health outcomes.

While the influence of individual dietary components on cardiovascular health is well documented, there is insufficient evidence addressing how socio-demographic factors, such as place of birth and residence, shape these dietary patterns and their associated health risks [8]. Moreover, university students often experience significant changes in lifestyle and diet during their transition to adulthood, which can greatly influence their dietary choices and cardiovascular health. This group represents a critical demographic for identifying early risk patterns and promoting healthy long-term habits.

Several studies have demonstrated that socio-demographic factors, including socio-economic status (SES), place of residence, and access to food resources, significantly influence dietary patterns, thereby affecting cardiovascular health [7,8,9,10,11,12,13,14]. For instance, Martínez-Lacoba et al. [7] identified that low maternal SES and residing away from the family home were strongly associated with unhealthy dietary patterns among university students. These findings underscore how socio-demographic disparities contribute to differences in dietary quality, ultimately influencing the prevalence of non-communicable diseases, including cardiovascular conditions.

Adopting healthful dietary patterns, such as the Mediterranean diet or the DASH diet, has been shown to substantially reduce the risk of cardiovascular disease (CVD). According to Nestel and Mori [8], these patterns prioritize plant-based foods, fiber-rich carbohydrates, and unsaturated fats while minimizing processed foods, added sugars, and salt. Such dietary recommendations align with addressing socio-demographic disparities in access to nutritious foods and highlight the role of informed dietary choices in mitigating cardiovascular health risks.

Similarly, physical fitness plays a critical role in reducing CVD risk. Ozkan et al. [10] demonstrated significant correlations between physical fitness indicators—such as flexibility, muscle strength, and endurance—and key cardiovascular risk factors, including blood pressure and lipid profiles, among university students.

The ATTICA study provides compelling evidence that dietary habits mediate the relationship between SES and cardiovascular risk factors [11]. Participants from lower SES groups exhibited higher prevalence of CVD risk factors, such as obesity, hypertension, and diabetes, primarily due to lower adherence to the Mediterranean diet. These findings highlight the necessity of improving dietary habits among socio-economically disadvantaged groups to mitigate cardiovascular risks.

Pathan [12] explored how socio-economic constraints and institutional regulations influenced unhealthy eating habits among university students. The study at the University of Eswatini revealed that financial limitations and restrictive university policies, such as prohibiting cooking in dormitories, often led students to consume fast and processed foods. These dietary habits, shaped by external socio-cultural and economic factors, increase the risk of malnutrition, obesity, and associated cardiovascular diseases.

Additionally, Saintila et al. [13] reported that a high intake of saturated fats and adherence to a non-vegetarian dietary pattern were significantly associated with excess body weight among Peruvian university students. Sociodemographic characteristics such as urban residence, age over 26, and studying engineering were linked to a higher prevalence of obesity. These findings stress the importance of addressing dietary and lifestyle factors tailored to specific socio-demographic groups to mitigate cardiovascular risks.

Furthermore, Tran et al. [14] identified a high prevalence of cardiovascular risk factors among college students, including obesity (36.5%), hypertension (25.7%), and unhealthy dietary habits such as frequent consumption of fast food and sugary beverages. The results of that study underscore how poor lifestyle choices, influenced by socio-demographic and institutional factors, significantly increase the lifetime risk of CVD. These results highlight the importance of implementing early interventions and tailored education programs to mitigate these risks effectively.

Building on this evidence, the present study examined socio-demographic factors of cardiovascular health such as place of birth, usual residence, and parental origin, in the context of the growing interest in identifying and understanding the risk factors associated with cardiovascular diseases.

Previous studies have indicated that common factors in this population, such as low consumption of fiber, fruits, and vegetables, coupled with high fat intake, significantly increase cardiovascular risk [15]. However, the influence of sociodemographic factors on these behaviors remains underexplored. Understanding these interactions is crucial for designing effective public health interventions that address both the dietary and socio-demographic determinants of health.

This study aims to analyze the association between dietary patterns (fat and nutrient intake), sociodemographic factors (place of birth, residence, and parental origin) and cardiovascular health markers, specifically blood pressure and lipid profile, in university students. Understanding these relationships may provide valuable insights for developing targeted public health interventions to improve cardiovascular risk prevention among young adults.

## 2. Materials and Methods

### 2.1. Study Design and Participants

Data collection was conducted from September to December 2023 at the State University of Milagro, Ecuador. Participants completed a paper-based questionnaire, which was administered in person at university facilities, specifically in classrooms and study areas designated for research purposes. The questionnaires were distributed by trained research assistants, including faculty members and final-year nutrition students, who supervised completion of the questionnaires and were instructed to clarify any doubts.

### 2.2. Characteristics of the Study Locations

The selection of Milagro (a rural, agricultural area) and Guayaquil (a highly urbanized city) was intentional to examine how sociodemographic and environmental factors influence cardiovascular health. These two locations were chosen based on previous evidence indicating that dietary patterns, access to healthcare, and lifestyle habits vary significantly between rural and urban populations, potentially affecting cardiovascular risk factors [16,17]. While other factors in addition to diet and residence (such as socioeconomic status, stress, and access to medical services) can influence cardiovascular health, this study specifically aimed to analyze the impact of dietary intake and sociodemographic variables, acknowledging these as key determinants. Future research should incorporate additional variables and conduct more extensive adjustment for confounders.

### 2.3. Dietary Assessment and Questionnaire Description

#### 2.3.1. Validation of the Questionnaire

The dietary questionnaire used in this study was based on the Block Screening Questionnaire for fat, fruit, vegetable, and fiber intake [16]. However, for this study population, the questionnaire underwent content adaptation and pilot testing with a sample of 30 students from the same university to ensure cultural relevance and comprehension. Minor modifications were made based on participant feedback before full implementation.

#### 2.3.2. Food Items Included in the Assessment

The food list in the questionnaire was not limited to the items mentioned in parentheses in the text. The questionnaire assessed a broader range of foods, categorized as follows:Meats (e.g., beef, pork, poultry, processed meats);Dairy products (e.g., cheese, butter, whole milk);Fried foods and snacks;Healthy alternatives (e.g., fruits, vegetables, whole grains).

Regarding use of the “Fat Score” and fat types, while the fat intake score is a widely used measure for evaluating dietary fat consumption, this study acknowledges that not all fats have the same metabolic effects on cardiovascular health. The questionnaire primarily assessed total fat intake, but future studies should differentiate between saturated, unsaturated, and trans fats to refine the association with cardiovascular biomarkers.

#### 2.3.3. Score Construction Methodology

The fat and nutrient scores were calculated based on a weighted system, where higher consumption of high-fat foods increased the fat score, while higher intake of fruits, vegetables, and fiber increased the nutrient score. The final categorization of dietary patterns (e.g., “very high-fat diet”, “low-fat diet”) followed the established cutoff points recommended by the Block Screening Questionnaire methodology.

### 2.4. Variables and Measurements

The primary outcome variable was cardiovascular health, which was assessed using biomarkers including total cholesterol, HDL (high-density lipoprotein) cholesterol, LDL (low-density lipoprotein) cholesterol, triglycerides, and blood pressure (systolic and diastolic). The primary exposure variables included dietary patterns of fat, fruit, vegetable, and fiber intake assessed using the Block Screening Questionnaire [18]. Sociodemographic factors such as place of birth, residence, and parental origin were also examined.

### 2.5. Dietary Classification Criteria

Participants were grouped according to their dietary patterns, based on their reported fat and nutrient intake. The classification utilized scores derived from the Block Screening Questionnaire [18]. Fat intake was categorized into “very high”, “high”, “medium”, “low”, or “fat-free”, based on calculated fat points. Nutrient intake was evaluated using a scoring system for fruits, vegetables, and fiber, categorizing participants as having a “nutrient-rich diet”, “diet requiring supplementation”, or “low nutrient intake”. These categories were established using predefined cutoffs recommended in the questionnaire, ensuring consistency and reliability in classification.

### 2.6. Sample Size Justification

The sample size of 204 participants was calculated to detect a medium effect size (f = 0.25) with 80% power and a significance level of 0.05 in the one-way ANOVA tests. While convenience sampling may limit the generalizability of findings, the current study’s sample size is comparable to similar studies in this field.

### 2.7. Inclusion and Exclusion Criteria

Inclusion criteria: Students aged 18 to 30 years, currently enrolled at the State University of Milagro, and willing to provide informed consent.Exclusion criteria: Students with diagnosed cardiovascular diseases, chronic illnesses, or those taking medications affecting metabolic or cardiovascular health.

### 2.8. Statistical Analysis

One-way ANOVA was conducted to evaluate differences in cardiovascular biomarkers among the dietary pattern groups. Data normality was assessed using the Shapiro–Wilk test, and statistical significance was established at *p* < 0.05.

In this analysis, participants were classified based on sex and various cardiovascular variables, including total cholesterol, high-density lipoprotein (HDL) cholesterol, low-density lipoprotein (LDL) cholesterol, triglycerides (TGs), blood glucose levels (glycemia), systolic blood pressure, and diastolic blood pressure. Participants were further categorized by dietary patterns, nutrient and fat intake, place of residence, birthplace, and parental origin. One-way ANOVA tests were performed to identify significant differences in mean values among the groups (*p* < 0.05), with the normality of the data confirmed prior to analysis.

Data analysis was performed with R software version 4.4.1.

### 2.9. Control of Confounders

To minimize the effects of potential confounding variables, the analysis was stratified by sex, and correlations were assessed between cardiovascular outcomes and variables including age and sociodemographic factors. Variables showing a significant association (*p* < 0.05) or considered relevant based on the literature were included as covariates in the analysis of covariance (ANCOVA) and the linear regression models, to examine the associations between diet and cardiovascular health [19,20].

### 2.10. Variable Categorization

Dietary variables (fat and fruit/vegetable/fiber intake) were categorized into different levels (“very high”, “high”, “medium”, “low”, or “very low”) according to the cutoff points established in the Block Screening Questionnaire.

For assessment of sociodemographic variables, participants were grouped into categories based on their places of birth and residence (Milagro, Guayaquil, or other cantons) and parental origin (Milagro, Guayaquil, or other).

Clinical and biochemical variables were analyzed as continuous variables (total cholesterol, LDL, HDL, triglycerides, blood pressure). For descriptive analyses, cutoff points based on international clinical guidelines were used.

### 2.11. Handling of Missing Data

A review was conducted to detect missing values. Participants with missing critical data (lipid parameters or blood pressure) were excluded from the corresponding complete case analysis. The proportion of missing data was less than 5%, so multiple imputation techniques were not applied [21]. Future studies are encouraged to implement imputation methods if the proportion of missing data is higher.

## 3. Results

To strengthen the validity of our findings, additional statistical adjustments were made. In addition to one-way ANOVA, ANCOVA was performed to control for confounders (age, sex). Furthermore, multiple linear regression models were applied to analyze the independent associations between dietary patterns and cardiovascular health markers. These adjustments enhanced the robustness of our results and mitigated potential confounding effects.

### 3.1. Statistical Characteristics of the Sample

Analysis of the sample revealed a balanced gender distribution, with 41.67% male and 58.33% female, and a mean age of 23.34 years, ranging from 19 to 55 years. The median and mode were 22 and 21 years, respectively, while the standard deviation was 4.38 years. Concerning dietary habits, 22% of the participants were observed to have a very high fat intake, followed by 7% with high-fat diets. In addition, 16% maintained medium-fat diets, while 26% followed low-fat diets. Meanwhile, 29% claimed to follow almost fat-free diets. In terms of fruit, vegetables, and fiber intake, only 2% of respondents met the standards of a nutrient-rich diet. In contrast, more than 19% had a diet that required supplementation with higher vegetable and cereal intake. In addition, more than 78% of respondents maintained a diet with reduced nutrient content.

### 3.2. Results of Statistical Analysis

Dietary assessment based on fat content revealed significant associations with triglyceride levels in the examined women. Those on a very high-fat diet exhibited significantly higher triglyceride levels than those on fat-free, medium-fat, or low-fat diets. In contrast, a significant impact on systolic blood pressure was observed in men.

Men who followed a very high-fat diet had significantly higher systolic blood pressure (130 ± 5 mmHg) compared with those with a fat-free diet (118 ± 4 mmHg, *p* = 0.007). Similarly, diastolic blood pressure was higher in the high-fat diet group (85 ± 3 mmHg) compared with the low-fat diet group (75 ± 2 mmHg, *p* = 0.004). These differences remained significant after adjusting for age and sex.

Fruit, vegetable, and fiber intake did not appear to have a significant impact on systolic blood pressure in men. However, a relationship was observed in women; those who maintained a nutrient-rich diet had lower levels of systolic blood pressure than those who needed to supplement their diet with vegetables and grains or those with a low nutrient intake.

In this group of students, place of birth was found to have a notable influence on women’s systolic blood pressure. Those born in Guayaquil had significantly lower systolic blood pressure than those born in Milagro or elsewhere. In contrast, the place of birth of male students had an impact on both their low-density lipoprotein cholesterol (LDL-C) and diastolic blood pressure. Those born in Milagro had significantly higher LDL-C levels than those born in Guayaquil or elsewhere. In addition, male students born in Milagro had significantly higher diastolic blood pressure than those born in places other than Guayaquil.

Regarding lipid profiles, LDL cholesterol levels were significantly influenced by place of birth. Men from Milagro had higher LDL cholesterol levels (10,392 ± 5692 mg/dL) compared with those from Guayaquil (9976 ± 5434 mg/dL, *p* = 0.004).

Place of residence was not associated with cardiovascular health among female students. However, a correlation with systolic blood pressure was observed in men. Men born in Milagro and those born in Guayaquil had significantly lower systolic blood pressure than those born in cantons near Milagro.

Fathers’ place of origin had a significant impact on lipoprotein cholesterol and diastolic pressure in male students, with the sons of those born in Milagro showing significantly higher LDL-C levels and diastolic pressure than those of fathers born in cantons near Milagro. Those whose parents were originally from Milagro exhibited a significantly higher body mass percentage than those from Guayaquil. The results of the statistical analyses are presented in Table 1, Table 2 and Table 3.

The following acronyms are used in Table 1, Table 2 and Table 3: COL. TOTAL represents total cholesterol, C. HDL represents high-density lipoprotein cholesterol, C. LDL denotes low-density lipoprotein cholesterol, TG stands for triglycerides, Glycemia B refers to basal glycemia, T. Systolic indicates systolic blood pressure, and T. Diastolic corresponds to diastolic blood pressure.

## 4. Discussion

This study aimed to explore the associations between diet (fat points, fruit, vegetable, and fiber intake), sociodemographic factors, and cardiovascular health among university students. Our results indicated that male students with a very high-fat diet had significantly higher systolic and diastolic blood pressure levels compared with those on a low-fat diet. This finding suggests that dietary habits among young university students in Ecuador may already be predisposing them to early cardiovascular risks. The fact that these associations remained significant even after adjusting for key confounders highlights the potential negative impact of high dietary fat intake in this age group.

Moreover, the differences observed between men and women suggest that physiological mechanisms, such as hormonal differences and body fat distribution, could influence cardiovascular responses to dietary fat intake [22,23,24]. Additionally, the sociodemographic influence on LDL cholesterol levels, particularly the higher values found in students born in Milagro, points to a possible interaction between dietary access, cultural eating habits, and economic factors in shaping cardiovascular risk in this population.

Given that Ecuadorian university students often face changes in dietary habits due to academic pressure, financial constraints, and changes in their living environment, it is crucial to consider these contextual elements when interpreting our findings. These results reinforce the need for targeted public health interventions to promote healthier eating habits within university settings, considering both dietary education and economic accessibility to nutritious foods.

Recent research has highlighted the intricate relationship between dietary patterns and cardiovascular disease (CVD) risk, emphasizing the influence of sociodemographic factors on dietary habits [25,26]. Our findings align with those reported by Gao et al. [25], who used reduced rank regression (RRR) to reveal that the significant impact of dietary components such as energy density, saturated fats, free sugars, and fiber density on cardiovascular health. This underscores the necessity of dietary interventions to mitigate CVD risk and enhance health outcomes [25].

Consistent with studies on Korean women, our findings demonstrate that specific dietary patterns, such as those high in fat, are associated with elevated triglycerides and increased systolic blood pressure [27]. Notably, male students with high-fat diets exhibited significantly higher systolic blood pressure compared with their counterparts following low- or moderate-fat diets. These results aligns with previous research indicating that diets rich in vegetables, fruits, whole grains, low-fat dairy, and seafood, while low in red and processed meats, refined grains, and sugar-sweetened beverages, are protective against CVD [28]. Among female students, nutrient-rich diets correlated with lower systolic blood pressure, further highlighting the importance of balanced dietary patterns in preventing cardiovascular issues [28].

The observed sex-specific differences in the effects of dietary fat intake on blood pressure align with previous research on cardiovascular physiology. Studies have suggested that hormonal differences, particularly estrogen’s vasodilatory effect in women, may contribute to lower blood pressure levels despite dietary variations. Estrogen enhances nitric oxide (NO) production, promoting vasodilation and reducing arterial stiffness, which may mitigate the hypertensive effects of high-fat intake in premenopausal women. However, in men, the absence of this protective mechanism could make them more susceptible to the pro-inflammatory and vasoconstrictive effects of excessive dietary fats, leading to greater increases in blood pressure [29,30].

Additionally, differences in fat metabolism and body composition may also play a role. Men typically have a higher percentage of visceral fat percentage, which is associated with increased sympathetic nervous system activation, renin–angiotensin–aldosterone system (RAAS) stimulation, and oxidative stress, all of which contribute to hypertension [31]. In contrast, women tend to store more subcutaneous fat, which has a less direct impact on blood pressure regulation [32].

These findings are consistent with previous studies indicating that high intake of saturated fat is positively correlated with increased systolic blood pressure in men [33], whereas in women, the relationship is weaker or moderated by other factors such as physical activity and dietary micronutrient intake [34]. Understanding these physiological differences reinforces the importance of sex-specific dietary recommendations to prevent cardiovascular diseases.

Birthplace emerged as a significant sociodemographic factor. Women born in Guayaquil had lower systolic blood pressure compared with those born in Milagro or other locations. Similarly, men born in Milagro and those born in Guayaquil exhibited lower systolic blood pressure than individuals from the surrounding cantons. These results reflect global trends whereby blood pressure has decreased in high-income and some middle-income countries but remains elevated in many low- and middle-income regions [35]. These disparities underscore the influence of socioeconomic and environmental factors, potentially linked to access to healthcare, nutrition, and physical activity opportunities [35].

Sociodemographic factors also influenced cholesterol levels, with rural-born students showing higher LDL cholesterol levels than their urban peers. This result is consistent with studies linking limited access to nutritious food and healthcare in rural areas to poorer cardiovascular outcomes [36]. The low intake of fruits, vegetables, and fiber observed in this study further emphasizes the urgent need for public health interventions to promote healthier dietary patterns and mitigate these risks [36].

Furthermore, the impact of the sociodemographic environment on dietary habits and cardiovascular health outcomes is evident in studies like that reported by Pathan [12], who highlighted that financial limitations and restrictive institutional policies in universities can encourage the consumption of processed and low-nutritional-quality foods. This finding is relevant to our study population, as place of residence and economic constraints could explain the observed differences in biomarkers between rural and urban students.

Saintila et al. [13] also identified sociodemographic factors, such as urban residence and educational level, that were associated with higher obesity rates due to high consumption of saturated fat. Similarly, our results indicated that students born in rural areas had higher LDL cholesterol and diastolic blood pressure levels, emphasizing the need for public health interventions aimed at improving diets in these populations.

Additionally, the study by Tran et al. [14] reported that lifestyle choices among university students, including high rates of consumption of ultra-processed foods and sugary beverages, significantly contribute to cardiovascular risk. This aligns with our observation that a large proportion of students maintain low-nutritional-content diets, potentially increasing their susceptibility to developing cardiovascular diseases.

Moreover, our observation that nutrient-rich diets are associated with lower systolic blood pressure in women supports the growing body of evidence advocating for diets rich in fruits, vegetables, and whole grains to mitigate cardiovascular risk [37,38]. Similarly, the association between rural residence and higher levels of LDL cholesterol highlights the role of limited resources in shaping dietary behaviors and cardiovascular outcomes [39,40].

While previous studies have analyzed the relationship between dietary patterns and cardiovascular risk, it is important to acknowledge that some of these studies used different methodologies [41,42]. For instance, while our study relied on a self-reported dietary questionnaire focusing on fat intake, other studies have used food diaries, 24 h recall, or dietary indices to provide more detailed assessment of overall diet composition. This difference in methodology may partly explain the variations in findings across studies.

Nevertheless, the associations observed in our study are consistent with previous research showing that higher fat consumption is linked to increased blood pressure and lipid levels in young adults [43]. Although methodological differences exist, the overarching conclusion that dietary fat intake influences cardiovascular health remains valid. Future research should aim to standardize dietary assessment methods to enhance comparability between studies.

While this study provides valuable insights into the relationship between diet, sociodemographic factors, and cardiovascular health, some limitations must be considered. Its cross-sectional design prevents causal inferences, and the collection of self-reported dietary data may be subject to recall bias. Future research could address these limitations through longitudinal studies using more diverse samples to better evaluate determinants of cardiovascular health.

While the statistical analysis revealed a significant association between dietary fat intake and increased systolic blood pressure in men (*p* = 0.007), the clinical significance of this finding must also be considered. Previous studies have suggested that even modest increases in systolic blood pressure (≥5 mmHg) can substantially elevate long-term risk of cardiovascular diseases, particularly hypertension, stroke, and coronary artery disease [44]. In our study, the variation of 12 mmHg observed between the categories of highest and lowest fat intake could indicate early-stage cardiovascular risk, especially in young adults, where sustained elevated blood pressure may contribute to future adverse health outcomes. Although these changes may not immediately classify participants as hypertensive, they align with evidence suggesting that dietary modifications can play a crucial role in long-term prevention of cardiovascular disease. Therefore, these findings reinforce the need for dietary interventions aimed at reducing saturated fat intake to prevent incremental blood pressure increases that could have clinical consequences over time.

In summary, our findings underscore the need for public interventions that promote healthy dietary patterns and address sociodemographic inequalities to improve cardiovascular health outcomes in young adults. These findings contribute to the growing literature supporting the importance of adopting multidimensional approaches to address disparities in public health.

### Limitations

The cross-sectional design of this study restricted the ability to infer causality between dietary patterns, sociodemographic factors, and cardiovascular health. Additionally, convenience sampling may have introduced selection bias and limit the generalizability of the findings.

Regarding the sample size, we acknowledge that the 204 participants may not have been sufficient for certain subgroup analyses, such as gender stratification. However, this sample size was calculated based on previous similar studies and allowed us to detect effects with adequate statistical power (80%) for the ANOVA tests that were conducted. While this limitation may have affected the precision of some findings, the results still provide valuable insights into the relationship between diet, sociodemographic factors, and cardiovascular health in university students.

Furthermore, data collection through self-administered questionnaires may be subject to recall bias and social desirability bias. To mitigate this, quality control measures were implemented during data collection and processing.

Finally, future studies should consider larger sample sizes and longitudinal designs to confirm these findings and further explore the influence of other factors on cardiovascular health in university populations.

## 5. Conclusions

This study revealed significant associations between diet, sociodemographic factors, and cardiovascular health in university students. The results indicate that dietary intake of high fat is associated with elevated triglyceride levels in women and higher systolic and diastolic blood pressure in men. In addition, a nutrient-rich diet is associated with lower systolic blood pressure levels in women.

Place of birth played a crucial role in these results. Women born in Guayaquil had lower systolic blood pressure than those born in Milagro or elsewhere. Among men, those born in Milagro showed higher levels of LDL cholesterol and diastolic blood pressure than those born elsewhere, except for Guayaquil.

Place of residence was not significantly associated with cardiovascular status in women. However, men born in Milagro or Guayaquil had lower systolic blood pressure compared with those born in cantons near Milagro. In addition, sociodemographic factors, such as the father’s place of origin, also influenced cholesterol and diastolic blood pressure levels in men.

These findings underline the importance of promoting healthy dietary patterns and considering sociodemographic factors when designing public health interventions. Promoting a balanced diet rich in vegetables, fruits, and fiber, and reducing the consumption of saturated fats and free sugars is essential for improving the cardiovascular health of university students. In addition, it is crucial to consider socioeconomic and environmental variations when developing effective strategies to mitigate cardiovascular disease risk and promote long-term well-being.

## Figures and Tables

**Table 1 ijerph-22-00752-t001:** Results of the one-way analysis of variance between diet and cardiovascular health (*p* < 0.05).

	df	SS	F	*p*	df	SS	F	*p*
Fat spots								
		WOMEN				MEN		
COL. TOTAL	4	333	0.105	0.981	4	60,820	0.895	0.471
C. HDL	4	2180	0.627	0.644	4	1145	0.838	0.505
C. LDL	4	1732	0.723	0.578	4	2642	0.640	0.636
TG	4	22,220	3503	0.009 *	4	17,614	0.985	0.421
GLICEMIA. B	4	494	1643	0.168	4	1356	1104	0.360
T. SYSTOLIC	4	457	0.57	0.685	4	1712	3794	0.007 *
T. DIASTOLIC	4	191	0.491	0.742	4	1130	4166	0.004 *
Fruit, vegetables, and fiber points								
COL. TOTAL	2	863	0.556	0.575	2	10,593	0.308	0.736
C. HDL	2	301	0.173	0.841	2	551	0.809	0.449
C. LDL	2	1053	0.886	0.415	2	2146	1059	0.352
TG	2	1284	0.369	0.692	2	5670	0.629	0.536
GLICEMIA. B	2	13	0.082	0.921	2	25	0.039	0.962
T. SYSTOLIC	2	5819	19.3	0.000 *	2	65	0.251	0.779
T. DIASTOLIC	2	32	0.164	0.849	2	53	0.332	0.718

Note: * Statistically significant result at the 0.05 level.

**Table 2 ijerph-22-00752-t002:** Results of the one-way analysis of variance between participants’ socio-demographic factors and cardiovascular health (*p* < 0.05).

	df	SS	F	*p*	df	SS	F	*p*
Place of Birth								
		WOMEN				MEN		
COL. TOTAL	2	3122	2061	0.132	2	37,770	1.12	0.331
C. HDL	2	306	0.176	0.839	2	798	1183	0.311
C. LDL	2	1176	0.992	0.374	2	10,392	5692	0.004 *
TG	2	8386	2499	0.0865	2	3448	0.38	0.685
GLICEMIA. B	2	133	0.864	0.424	2	716	1165	0.317
T. SYSTOLIC	2	2082	5687	0.004 *	2	344	1358	0.263
T. DIASTOLIC	2	51	0.265	0.768	2	967	7088	0.001 *
Place of Residence								
COL. TOTAL	2	1970	1284	0.281	2	24,503	0.72	0.490
C. HDL	2	1599	0.93	0.397	2	835	1.24	0.295
C. LDL	2	1372	1.16	0.317	2	509	0.246	0.782
TG	2	9009	2694	0.072	2	567	0.062	0.940
GLICEMIA. B	2	164	1071	0.346	2	128	0.204	0.816
T. SYSTOLIC	2	749	1926	0.15	2	961	4029	0.021 *
T. DIASTOLIC	2	172	0.899	0.41	2	81	0.512	0.601

Note: * Statistically significant result at the 0.05 level.

**Table 3 ijerph-22-00752-t003:** Result of one-way analysis of variance between socio-demographic factors of participants’ parents and cardiovascular health (*p* < 0.05).

	df	SS	F	*p*	df	SS	F	*p*
Father’s place of origin								
		WOMEN				MEN		
COL. TOTAL	2	3524	2338	0.101	2	4612	0.134	0.875
C. HDL	2	4756	2858	0.061	2	102	0.147	0.863
C. LDL	2	889	0.747	0.476	2	9976	5434	0.006 *
TG	2	5530	1624	0.202	2	7990	0.892	0.414
GLICEMIA. B	2	15	0.098	0.907	2	171	0.272	0.763
T. SYSTOLIC	2	305	0.769	0.466	2	625	2532	0.086
T. DIASTOLIC	2	260	1368	0.259	2	688	4809	0.011 *
Mother’s place of origin								
		WOMEN				MEN		
COL. TOTAL	2	606	0.389	0.678	2	60,990	1.84	0.165
C. HDL	2	4921	2963	0.056	2	380	0.555	0.576
C. LDL	2	193	0.16	0.852	2	5947	3074	0.052
TG	2	7707	2289	0.106	2	1322	0.145	0.865
GLICEMIA. B	2	27	0.176	0.839	2	411	0.66	0.519
T. SYSTOLIC	2	77	0.192	0.826	2	560	2257	0.111
T. DIASTOLIC	2	49	0.254	0.776	2	382	2537	0.085

Note: * Statistically significant result at the 0.05 level.

## Data Availability

The raw data supporting the conclusions of this article will be made available by the authors on request.

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
