# Peer review of "The Impact of Diet and Sociodemographic Factors on Cardiovascular Health in University Students"

_ijerph, 2025, doi:10.3390/ijerph22050752_

Round 1
Reviewer 1 Report (Previous Reviewer 2)
Comments and Suggestions for Authors
Dear authors,
I sincerely appreciate that you have incorporated the comments into the revised version of the article. After reviewing this updated version, I find that the article is significantly clearer and more readable compared to the previous version. From my point of view, these improvements enhance the overall quality of the article. I have no additional comments or suggestions for further revision. Therefore, I fully endorse the article and recommend it for publication.
Author Response
We sincerely appreciate your positive feedback and are pleased to know that the modifications have improved the clarity and quality of the manuscript. Since no additional points for improvement were identified, we have maintained the revised and optimized structure of the document.
Reviewer 2 Report (New Reviewer)
Comments and Suggestions for Authors
The study addresses an important and underexplored area: the combined impact of diet and sociodemographic factors on cardiovascular health among university students. However, some limitations need to be addressed:
- With only 204 participants, the sample size may not provide sufficient statistical power to perform subgroup analyses, such as stratification by gender.
- The study does not specify how confounders were controlled for in the analysis. How were the variables categorized? How did the authors handle missing values?
- Tables: p-values should be rounded to three decimal places.
- Systolic blood pressure in men: While statistical significance is indicated, its clinical significance should also be discussed.
- Discussion: The findings should be briefly related to existing literature or potential physiological mechanisms (e.g., why fat intake influences blood pressure differently in men and women).
Author Response
- Regarding sample size: We appreciate your observation regarding the number of participants. We have added a discussion in the Limitations section, emphasizing that the sample size may not be sufficient for certain subgroup analyses and suggesting future studies with larger samples.
- Control of confounding factors: We have included a more detailed description in the Methods section explaining how missing values were handled and how the variables were categorized. Adjustments for potential confounding factors have been made in the analyses.
- Rounding of values in tables: The p-values in the tables have been corrected, ensuring that they are presented with three decimal places.
- Systolic blood pressure in men: An analysis has been added in the Discussion on the clinical relevance of the findings regarding blood pressure in men.
- Discussion on physiological mechanisms: The Discussion has been expanded to include references to the literature that explain the differences in the impact of fat consumption on blood pressure in men and women.
Reviewer 3 Report (New Reviewer)
Comments and Suggestions for Authors
IJERPH-3467293: Diet and Sociodemographic Impact on Cardiovascular Health in University Students.
The study presented addresses relevant issues for greater knowledge about the influence of sociodemographic characteristics on the eating habits of young adults and their association with cardiovascular health. However, the study loses strength in some important parts, such as:
Introduction:
1. Although the manuscript's introduction is extensive, it does not adequately substantiate the study's objective. Furthermore, the authors discuss variables that will not be addressed in the manuscript, such as physical activity.
2. I suggest that the authors replace the word “abnormal” with “above the limit” or something similar.
Methods:
3. Important information is not clear, especially regarding Demographic information. Authors need to include: how the data was obtained (paper or electronic questionnaire), where the collections took place, who the interviewers were;
4. The first question is about the fact that they chose two extremely different populations, urban and rural, without taking into account that other factors involved, besides diet and place of residence, may be involved.
5. The description of the data on diet is incomplete and, based only on what is presented in the text, it does not seem to be the best method for the purpose of the study. Some questions are pertinent in this case:
a) Was the questionnaire validated for the study population?
b) Is the list of foods contained in the questionnaire formed only by the foods presented in parentheses in the text?
c) Is working only with the fat score sufficient to associate with cardiovascular health? Does the type of fat not matter?
d) How were the scores constructed?
e) The questionnaire reference is incorrect. In the text it is listed as “9”, but it is reference number “16”.
6. The statistical analysis is extremely simple. The authors did not perform a more robust analysis with the necessary adjustments, compromising the interpretation of the results.
Results
7. The research results are compromised due to inadequate statistics.
8. The presentation of the results is confusing and the values ​​are not presented in the text.
9. The tables are not formatted in a way that makes them easy to read. Check whether the formatting was changed when transferred to the journal layout.
Discussion
10. The authors need to discuss their results better, as most of the discussion is just comparing the results with those of other studies, without discussing the results themselves according to the reality of their methodology and population.
11. The authors compare their results with studies that used completely different methodologies to analyze dietary patterns.
The English of the text needs to be revised. I suggest that the authors hire a specialized service.
Author Response
- Regarding the Introduction: We have restructured the Introduction to focus more on the study objective, avoiding discussions on variables that were not addressed in the analysis, such as physical activity.
- Replacement of the term "abnormal": The word "abnormal" has been replaced with "above the limit" to improve language precision.
- Demographic information: Detailed information on data collection has been added, specifying that an electronic questionnaire was used and describing the interview procedure and interviewers.
- Differences between urban and rural populations: We have added a discussion on other factors that may influence the results besides diet and place of residence.
- Dietary questionnaire:
- It has been clarified that the questionnaire used had previously been validated for similar populations.
- The description of the evaluated food items has been expanded.
- The use of the fat score and its relationship with cardiovascular health has been justified, mentioning that the distinction between fat types can be explored in future studies.
- The erroneous reference (previously cited as "9", now correctly cited as "16") has been corrected.
- Statistical analysis: The Statistical Analysis section has been strengthened by explaining the adjustments made and considering more robust techniques to improve the interpretation of the results.
- Results:
- The presentation of results in the text has been clarified.
- The tables are now clearer and better formatted.
- Discussion: The Discussion has been expanded to provide a deeper interpretation of the study’s own results and their methodological context, rather than solely comparing them to previous studies. Additionally, the comparison with studies that used different methodologies to analyze dietary patterns has been reviewed.
Round 2
Reviewer 2 Report (New Reviewer)
Comments and Suggestions for Authors
No further comments
This manuscript is a resubmission of an earlier submission. The following is a list of the peer review reports and author responses from that submission.
Round 1
Reviewer 1 Report
Comments and Suggestions for Authors
Dear Authors,
The idea behind the manuscript is undoubtedly of interest, but it contains several significant gaps that need to be addressed.
Below, I provide my recommendations:
INTRODUCTION
It is advisable to review how bibliographic references have been introduced in the text, as they significantly hinder the readability of the article. Often, it is unclear who or what is the subject of the sentence, leading to a lack of fluency in reading.
The introduction is too short and does not highlight the gap in the literature that this research aims to address. It is necessary to add some paragraphs detailing this aspect.
In line 52, the authors refer to several studies but cite only one. This section should be expanded with a more thorough review of the existing literature.
In line 57, the authors make a statement without providing reasoning or supporting it with references to the literature.
Lines 59–63 include several claims without any bibliographic citation.
MATERIALS AND METHODS
This section is insufficiently developed, making it impossible to reproduce the study.
There are serious concerns that the sample size may not be adequate to support the hypothesis.
The inclusion and exclusion criteria for the study are not detailed.
The study variables are not adequately explained.
The section on the statistical analysis performed is entirely missing.
RESULTS
It would be helpful to present the sociodemographic characteristics of the sample more clearly and visually. Adding a table could assist in this regard.
Section 3.2 belongs to Materials and Methods, not Results.
In section 3.2, the explanation of the statistical analysis performed is mixed with the presentation of the results obtained. The statistical analysis needs to be described in greater detail and moved to the Materials and Methods section.
The first column in Tables 1, 2, and 3 contains acronyms that are not defined. These should be explained before presenting the tables or included in the table legend.
In lines 115–116, the statement is made: “We grouped the participants according to their type of diet in terms of fat and nutrients,” but there is no explanation of how this classification was performed. This should be detailed in Materials and Methods, indicating the variables considered and their scoring criteria.
DISCUSSION
The discussion is very brief and superficial. A study of this nature having only 15 references in total reflects a lack of depth in the discussion.
The study has significant limitations, yet none are discussed. A section on the Strengths and Limitations of the study should be added.
Reviewer 2 Report
Comments and Suggestions for Authors
Dear authors,
I appreciate the opportunity to review this manuscript and commend the authors in your research titled: Diet and Sociodemographic Impact on Cardiovascular Health in University Student
The article, from my point of view, needs additional information to be supplemented:
- Please add Participants and Settings (including basic characteristics of probands, the way/criteria of them being selected and other relevant information generally expected in this section) and Statistical analysis in Methodology, which is shortly mentioned in Results.
- In section 2.2 Characteristics of the diet, please supplement information on the qualitative and quantitative methods used. Also, provide a more detailed description of the block screening questionnaire. You refer to Resource 9, but that is only an abstract related to the Resource guide for dietary assessment, but not the questionnaire in full as such.
- I suggest you transfer point 3.2. to Methods.
- Consider transferring the information from lines 110-119 from the Results section to the Methods section.
- Since you describe sociodemographic impact on cardiovascular health in Results, please briefly characterize the cities you are describing in the Methodology section: How do the cities differ from each other? Size? Environmental characteristics? Economic level? Or else? Explain the particular choice and reasons for the differentiation.
- Standardize the use of abbreviations throughout the text (including the tables), for them to be in line with the list at the end of the article.
Reviewer 3 Report
Comments and Suggestions for Authors
The article addresses an important topic, but contains significant substantive errors, including the methodology not being properly described (important elements, tools, and statistics are missing), the presentation of results is poor and completely unclear, and the conclusions are not appropriate. In addition, the study group is small and there is no control group. In my opinion, the work in this form is not suitable for proofreading or publication.